# The Discontinuous Elevational Distribution of an Ungulate at the Regional Scale: Implications for Speciation and Conservation

**DOI:** 10.3390/ani11123565

**Published:** 2021-12-15

**Authors:** Kun Tan, De-Pin Li, Na Li, Yi-Hao Fang, Yan-Peng Li, Wen Xiao

**Affiliations:** 1Institute of Eastern-Himalaya Biodiversity Research, Dali University, Dali 671003, China; tank@eastern-himalaya.cn (K.T.); lidp@eastern-himalaya.cn (D.-P.L.); Lin@eastern-himalaya.cn (N.L.); fangyh@eastern-himalaya.cn (Y.-H.F.); liyp@eastern-himalaya.cn (Y.-P.L.); 2The Key Laboratory of Yunnan Education Department on Er’hai Catchment Conservation and Sustainable Development, Dali 671003, China; 3Collaborative Innovation Center for Biodiversity and Conservation in the Three Parallel Rivers Region of China, Dali 671003, China

**Keywords:** elevational distribution, camera trap, blue sheep, invasive species, differentiation

## Abstract

**Simple Summary:**

The Himalaya blue sheep (*Pseudois nayaur*) in Baima Snow Mountain is found to exhibit a distinctive bimodal distribution along elevation gradient contrasting the unimodal distributions of most species. The first distributional peak of Himalaya blue sheep was in scree habitat around 4100 m a.s.l., while the second peak was in the dry-hot valley around 2600 m. Geographic separation from the original population along elevation suggested that population at lower elevation could be a separate species ecologically, or a taxa ongoing differentiation. Invasive species *Opuntia ficus-indica*, which colonized the region six hundred years ago, may have formed new foraging niche to support population at lower elevation. Our results suggested conservation measures should pay attention to taxa ongoing differentiation, and consider the possible active effect of biological invasion.

**Abstract:**

The elevational range where montane species live is a key factor of spatial niche partitioning, because the limits of such ranges are influenced by interspecies interaction, abiotic stress, and dispersal barriers. At the regional scale, unimodal distributions of single species along the elevation gradient have often been reported, while discontinuous patterns, such as bimodal distributions, and potential ecological implications have been rarely discussed. Here, we used extensive camera trap records to reveal the elevation distribution of Himalaya blue sheep (*Pseudois nayaur*) and its co-existence with other ground animal communities along a slope of Baima Snow Mountain, southwest China. The results show that Himalaya blue sheep exhibited a distinctive bimodal distribution along the elevation gradient contrasting the unimodal distributions found for the other ungulates in Baima snow mountain. A first distributional peak was represented by a population habituating in scree habitat around 4100 m, and a second peak was found in the dry-hot valley around 2600 m. The two distinct populations co-existed with disparate animal communities and these assemblages were similar both in the dry and rainy seasons. The extremely low abundance of blue sheep observed in the densely forested belt at mid-elevation indicates that vegetation rather than temperature is responsible for such segregation. The low-elevation population relied highly on *Opuntia ficus-indica*, an invasive cactus species that colonized the region six hundred years ago, as food resource. Being the only animal that developed a strategy to feed on this spiky plant, we suggest invasive species may have formed new foraging niche to support blue sheep population in lower elevation hot-dry river valleys, resulting in the geographic separation from the original population and a potential morphological differentiation, as recorded. These findings emphasize the important conservation values of role of ecological functions to identify different taxa, and conservation values of apparent similar species of different ecological functions.

## 1. Introduction

Understanding the mechanisms underlying the formation of vertical niche partitioning and the elevational distribution of montane biodiversity is a critical aspect in ecology and biogeography, providing basic references for conservation [1,2]. Both the abiotic environment, such as temperature, precipitation, and elevation, and biotic factors, such as population abundance, body mass, niche breath, and the interactions between species (predation, competition) influence and limit their distribution pattern [2,3,4,5,6,7,8]. Elevational gradients are also related to phylogenetic relationships and are used as a niche proxy in taxonomy [9]. The isolation of gene flow following spatial segregation is a critical driver of speciation. At the regional scale, single species usually exhibit a unimodal distribution along the elevation [6,7], such as ungulates in the Swiss Alps [10] and the Sikkim Himalaya [11], or burying beetles in Taiwan [1]. However, discontinuous elevational distributions of species in mountainous environments have rarely been reported and their potential ecological consequences are poorly discussed.

The bharal, also known as blue sheep (*Pseudois nayaur*), is a mountain ungulate endemic to the Pan-Himalaya area [12]. Most of the blue sheep populations in the Trans-Himalayas region live on high elevation mountain scree, avoiding habitat below 4000 m [13], while populations in the Yangze Gorge, southwest China, once considered as a full species, were also found below the forest belt [14]. A long-standing controversy exists about the taxonomical status of the low elevation populations, once named dwarf blue sheep (*Pseudois schaeferi*) [14,15]. Mitochondrial DNA and microsatellite markers were used to clarify the evolutional relationship of a geographically isolated population, but controversial conclusions on the taxa were proposed [14]. However, most of the recent studies failed to distinguish dwarf blue sheep in genetics from other populations living in the Hengduan Mountain chain.

Blue sheep prefer open vegetation [16] and avoid densely forested patches [17]. The low elevation river valleys of the Hengduan mountains area, such as the Jinsha and Min rivers, are characterized by a typical arid climate caused by the foehn effect, and vegetation dominated by grassland and shrubland [18]. It was found that an invasive cactus, *Opuntia ficus-indica*, was the main food resource consumed by blue sheep of the Jinsha valley on Baima Snow Mountain, especially during the winter season [19]. Based on this evidence, we hypothesize that the environment with scarce tree cover where a novel food resource is provided by the colonizing invasive plant, maintains the population in low elevation habitat, separating it completely from the population living at high elevation by the unsuitable mid-elevation forest belt.

Here, we survey the distribution of blue sheep and other ground animals by installing camera traps in different vegetation types distributed between 2000 and 4700 m. on Baima Snow Mountain area (BSM), southwest China. In this region, several montane species co-exist, including five additional ungulate species, namely, alpine musk deer (*Moschus chrysogaster*), dwarf musk deer (*Moschus berezovskii*), Chinese goral (*Naemorhedus griseus*), Chinese serow (*Capricornis milneedwardsii*), and tufted deer (*Elaphodus cephalophus*) co-exist. With the collected records, we assess the distribution patterns of the identified species and analyze the community co-existence of blue sheep in different vegetation types and elevations. We also review the elevational range of blue sheep species across other regions where the blue sheep occur. We predict that the blue sheep of BSM will show a particularly discontinued distribution along the elevation owing to the presence of suitable but divergent habitat at both low and high elevation, separated by an unsuitable densely vegetated belt. We also expect differences in their co-existence with other animal communities occurring in divergent habitats.

## 2. Methods

### 2.1. Study Area

The study was conducted on the eastern slope of southern Baima Snow Mountain National Nature Reserve (BSM, 27°47′–28°36′ N, 98°57′–99°21′ E, Figure 1), located in Deqing County, Diqing Tibetan Autonomous Prefecture in Yunnan, within the extensive Hengduan Mountains range. BSM stands between the Yangzi River in the east and the Lancang River in the west, with elevation ranging from 1950 to 5429 m. The nature reserve belongs to one of the world’s major biodiversity hotspots, known as the Mountains of South-Central China [20]. The marked seasonal changes in temperature and precipitation are governed by the south-west monsoon from the Indian Ocean. The interaction of the monsoon and the sharp slope shapes the vertical zonation of climate. The annual average temperature ranges from 16.5 °C (2000 m) to −7.4 °C (5500 m), and the annual precipitation ranges from 200 mm (2000 m) to 1000 mm (4500 m) from low to high elevation in our study site. The vegetation types change dramatically along with altitude. Rini Mountain, the low conical hill in the southeast of the study region separated from the main range by a low saddle (Figure 1), peaks at 3650 m and is dominated by shrubs including *O. ficus-indica*. Broad-leaved (dominated by *Quercus aliena var. acutiserrata, Acerspp., Betula albosinensis, Betula platyphylla,* and *Populus davidiana*) and coniferous forests (dominated by *Pinus yunnanensis, Pinus armandii, Tsuga dumosa, Pinus densata, Picea likiangensis, Abies georgei, and Platycladu sorientalis*) are found between 2600 and 4200 m, while alpine shrubs (dominated by *Rhododendron impeditum, Rhododendron pingianum, Rhododendron russatum, and Juniperus squamata*) and scree are usually above 4000 m [21].

### 2.2. Data Collection

#### 2.2.1. Camera Traps

We installed 108 infrared-triggered camera traps (Ltl Acorn 6310) stations (Figure 1), active 24/24 h from November 2015 to June 2016. We chose the locations according to dominant vegetation types, determined in 10 × 10 m plots centered on each trap station: valley shrub, broad-leaved forest, coniferous forest, meadow, alpine shrub, and alpine scree (Table 1). The cameras were positioned at the height of 0.5 to 1m from the ground and the delay period between photographs was set to 0 s.

#### 2.2.2. Data Analysis

Camera days for each camera were counted from the date of deployment until the date of the last photo. Consecutive captures for the same individuals or species (when individuals could not be identified) at the same station were considered as independent observations (IO) when separated by at least a one-hour interval. The relative abundance index (RAI) of each species was defined as the number of IO per 100 camera days [22]. 

The RAI of each ground species in each camera trap were plotted along the elevation gradient to reveal their vertical distribution. The ungulate community structure and composition in each station with different elevation and vegetation type were assessed using Nonmetric Multi-dimensional Scaling ordination (NMDS) [23]. The NMDS ordination was based on the Bray-Curtis dissimilarity metric of a species × sample-points abundance matrix. NMDS analyses were performed using 20 random starting configurations, and minimal accepted stress levels (a measure of goodness of fit) close to or <0.2 were used to determine the dimensions of ordination. The separation between low and high elevation populations of blue sheep was set at 3500 m. To evaluate whether the resulting elevation distribution pattern is stable across seasons, the elevation distribution plots and NMDS analyses were conducted both for the dry (October to March) and rainy seasons (April to June) separately, according to the temperature and rainfall pattern. We only included stations with more than 30 camera days and using RAI of identified species. The NMDS ordination was performed in the R statistics programming environment with the package “vegan” [23]. 

### 2.3. Review of Published Elevation Ranges of Blue Sheep

We searched the Google Scholar database with the key words “Blue sheep” or “Bharal” or “*Pseudois nayaur*” or “*Pseudois schaeferi*”. Only the publications that provided sufficient elevation information of the species were included, and redundant records (e.g., same region) were excluded.

## 3. Results 

A total of 60 species were identified from the collected records, including 41 birds and 19 mammals (Appendix A). According to our protocol, 97 stations had more than 30 camera days and thereby qualified for the distribution and community analyses. The IO of blue sheep was 662 in 39 stations, of which 33 in 11 stations were in the alpine shrub and scree. The IO for the other ungulate species were: Chinese goral 895 in 43 stations, Chinese serow 169 in 37 stations, tufted deer 191 in 22 stations, dwarf musk deer 77 in 15 stations, and alpine musk deer 6 in 3 stations.

### 3.1. The Distribution along Elevation

The RAI of blue sheep were higher in the valley shrubs, followed by alpine screes and meadows, while those in forest and alpine shrub were much lower (Table 1). Along the elevation gradient, the RAI of blue sheep showed a bimodal distribution, with a first peak in the habitats located around 2600 m and a second peak around 4100 m (Figure 2).

The Chinese goral shared the habitat with the blue sheep population at low elevation, showing a unimodal elevational distribution. Chinese serow and Tufted deer showed similar patterns, with their peaks were around 3000 to 4000 m, while the Alpine musk deer was mostly observed at a slightly higher elevation. We excluded alpine musk deer because of the low data collected. Species showed a similar pattern in dry and rainy seasons (Appendix A).

### 3.2. Patterns of Community Assemblage

The NMDS community ordination analysis for the full data-set produced a two-dimensional solution (stress = 0.06). The NMDS plots showed separated community compositions according to vegetation types (Figure 3). In particular, the community occurring in valley shrub habitats were distinct from other vegetation types, while the communities found in broad-leaved and coniferous forests were more similar. The low elevation population of blue sheep and Chinese goral co-existed in valley shrubs. Instead, the Chinese serow, alpine musk deer, dwarf musk deer, and tufted deer mainly occurred in forested patches. Instead, the high elevation blue sheep population occurred in alpine scree and shrubs, not sharing the habitat with other ungulates. The community structure remained consistence between the dry and rainy seasons.

### 3.3. Elevation Range of Sheep across Regions

A total of 10 studies focusing on blue sheep distribution in four regions were collected (Appendix A). Most of the blue sheep in the Trans-Himalaya and Tibet region occurred in areas located above 4000 m, but those in Yak Kharka and upper Mustang in Nepal were also found between 3000 and 4000 m [24]; The population in the Alxa area, which is located at the highest latitude, occupied habitats between 1500 and 4000 m [25,26]. In Batang County [15] and Wanglang Nature Reserve [27], which lie within the Hengduan mountains, blue sheep populations occurred between 3000 to 4000 m, below the timberline.

## 4. Discussion

The results of elevational distribution of blue sheep populations in several regions and other ungulates in BSM showed a general tendency to a unimodal distribution along the elevation gradient. However, our focus blue sheep population in BSM presented a peculiar bimodal distribution, with peaks at lower and higher elevations and a void at mid-elevation, where densely forested areas are found. This discontinuous elevational distribution is consistent with our hypothesis. Although the study was not conducted throughout a whole year, we were able to reveal a similar pattern in both dry and rainy season. Such kind of distribution pattern has also been reported in Batang, located at the west bank of the Jinsha river (a section of the Yangzi) [15]. The absence of blue sheep in the 3500 to 4000 m elevation range in both seasons and a very low abundance found in forests indicate that vegetation rather than temperature explains such discontinuity.

In general, invasive species have a negative impact on the biodiversity of the affected areas, creating niche displacement, competitive exclusion, or predation [28,29]. However, in some cases, they may provide foraging or breeding resources for other native species. For example, rapid evolution of native species as a response to invasive species, such as adapted feeding preferences or breeding behaviors, have been observed [30,31], and new taxa can form through hybridization and introgression [32]. Theoretically, those niche opportunities provided by invasive species can potentially drive the ecological divergence within a species, leading to species formation, as the ecological speciation concept suggests. The dominant diet of blue sheep populations in Trans-Himalaya and Helan mountain is graminoid and herbs [33], but the Helan mountain population also consume tree and shrub species [17]. The distinctive diet of the blue sheep population in dry-hot valley, characterized by invasive species [19], and the unique elevational distribution found there, suggest that invasive food resources may also be important drivers of blue sheep dispersion into the low valley habitats resulting in geographical separation of the former population. The spread of *Cactus* in Yunnan Province can be traced to more than six hundred years ago when *Cactus* was used as ornamental flowers or fences [34]. *Cactus* is a widely used fodder in Latin America and Africa as a buffer food lacking fiber and nitrogen [35]. Grove [36] once pointed out that the body size of dwarf blue sheep was significantly smaller than those of other species. The low-quality food and the warmer weather in low elevation valleys may be the reasons for the smaller body size. Dispersing to low elevation means the sheep had to co-exist with other herbivore competitors, as shown in the present study by the Chinese goral. To feed on these spiky plants, the blue sheep population developed a specific foraging strategy consisting in the use of their broad curved horns to remove the spines from *O. ficus-indica’* [19]. The horns of gorals, instead, are too short to adopt this tactic efficiently. Therefore, we propose that this resulting foraging partitioning alleviates the competition between sheep and gorals in low-elevation hot-dry valleys, making the habitat shift for blue sheep possible and allowing a sustainable co-existence with the gorals.

Biological species, defined by their phylogenetic relationships [37], are usually considered as the unit in conservation biology. Although it has been long recognized that variety of ecological functions exists within one species [38,39], the functional diversity within a species in an ecosystem is rarely discussed in conservation. The low elevation blue sheep population in our study, which represents the second most abundant species in the warm-hot valley environment, appears to fulfill different functions from the high elevation population. Their distinguished feeding habits may have an important role in regulating the demography and distribution of the invasive *Cactus*. Most of the previous studies based on molecular biology argued that the low-elevation population was not an isolated species, leading to its removal from the IUCN Red List of Threatened Species. However, when considering their different ecological functions and their geographically separated, population at lower elevation should be treated as a separate species. Moreover, speciation is a long-time process, conservation perspective should consider on those taxa under ongoing ecological differentiation.

## 5. Conclusions

The discontinuous distribution along the elevation gradient of the Himalaya blue sheep (*Pseudois nayaur*) results from climatic, food resources, and plant invasion processes. Invasive species do not always decrease regional biodiversity as generally supposed, but can promote speciation by providing new foraging niches. Conservation biology should put emphasis on those taxa owning functional differentiation.

## Figures and Tables

**Figure 1 animals-11-03565-f001:**
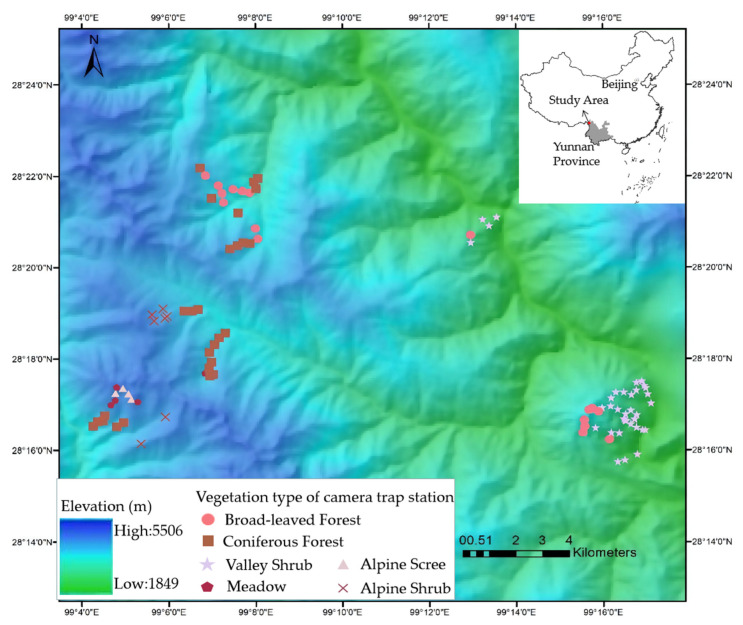
Study area and location camera trap stations.

**Figure 2 animals-11-03565-f002:**
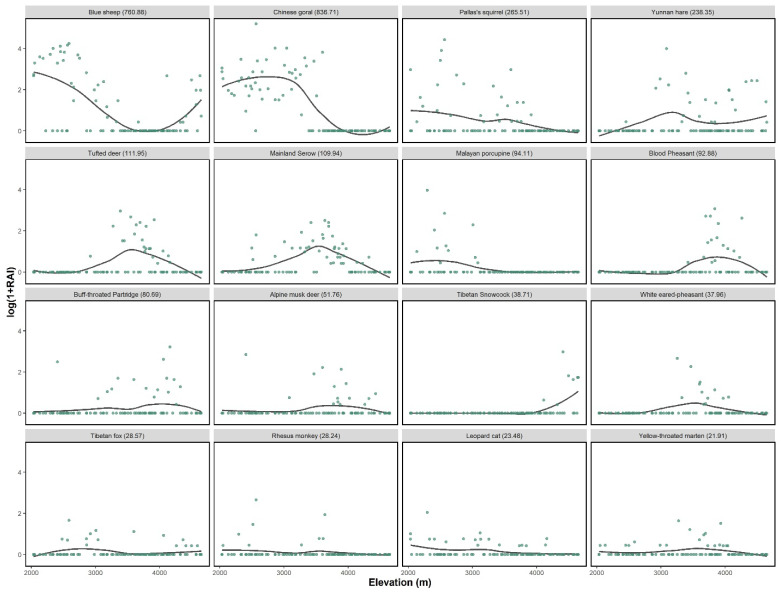
Relative abundance index (RAI, log scale) of different species against elevation in Baima Snow Mountain National Nature Reserve. The total RAI was stated in the bracket.

**Figure 3 animals-11-03565-f003:**
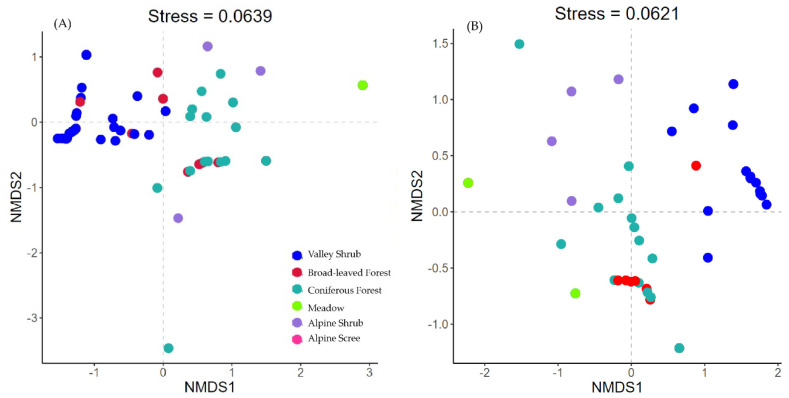
NMDS plots of community in different vegetation types and the projection of ungulates in (**A**) dry season and (**B**) rainy season. Each dot represents a camera trap station: the closer the two points are, the more similar their community composition. The different colors represent the vegetation types. The axes of the NMDS present the ordination dimensions that distinguish the communities. NMDS plots are presented based on Bray-Curtis dissimilarity of community composition. The stress value is a measure of the goodness of fit of the resultant ordination feature space as compared to the original data feature space, where zero is equal to perfect representation.

**Table 1 animals-11-03565-t001:** Number of stations and camera days in each vegetation type and relative abundance index of (RAI) of blue sheep.

Vegetation Type	Elevation Range (m)	Station Num.	Camera Days	RAI (mean + SD)
Valley Shrub	2030~3184	36	4428	19.91 ± 21.15
Broad-leaved Forest	2660~4053	15	1936	0.42 ± 1.10
Coniferous Forest	3275~4260	30	4448	0.52 ± 2.60
Meadow	3925~4649	6	894	3.10 ± 4.44
Alpine Shrub	3975~4429	6	916	0.14 ± 0.25
Alpine Scree	4539~4639	4	642	6.47 ± 5.51

## Data Availability

The data presented in this study are available on request from the corresponding author.

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
