# Peer review of "The Discontinuous Elevational Distribution of an Ungulate at the Regional Scale: Implications for Speciation and Conservation"

_animals, 2021, doi:10.3390/ani11123565_

Round 1

Reviewer 1 Report

In this study the authors use camera traps to evaluate patterns of narrow elevational distribution of Himalaya blue sheep Pseudois nayaur and other species across an elevational range of 2000 to 4500 m in the Baima Snow Mountains, SW China. They also include a brief review of published and anecdotal records of blue sheep elevational distribution across its global range. It is not easy to conduct camera trap surveys in these montane environments and the authors deserve some credit for their fieldwork efforts! Overall the manuscript is brief, and the analyses is somewhat limited. I have some concerns regarding the content.

Introduction, Aim and Objectives: There is simply not enough background to the main theme of the study – which are the factors that influence species elevational ranges/distributional patterns at different scales. A large volume of studies on the factors driving patterns of species elevational ranges in montane environments (for other taxa, such as birds, plants etc), particularly from the Andes, have not been consulted. A multitude of papers have examined how multiple interacting factors influence the patterns of narrow range boundaries, with both biotic and abiotic factors operating and local and landscape scales – these influence the persistence of species  populations. At the local scale, local climate, ecotones, competition and habitat structure/complexity are more important, whereas at the landscape level, biogeographic factors such as area of habitat, degree of isolation and climatic gradients are more important. None of this is clearly outlined in their current Introduction (or Discussion), thus this section has a significant literature gap. In addition, their main prediction is also somewhat redundant as the discontinuous distribution is already stated earlier in the Introduction. A more valuable study would be to examine distributional patterns of their five target ungulates, or all the species recorded (shown in the supplementary table) and to broaden not just the number of species sampled, but also expand the analyses (see below). This way, the authors could adopt a more question-based approach (i.e. present research questions as specific objectives) and examine how, not just species populations are distributed, but also patterns of species diversity, perhaps even patterns of beta-diversity across the narrow elevational range examined.

Study site description: This is too brief and there is not enough detail regarding the elevational gradient, and turnover of habitats i.e. presence/absence of sharp/distinctive ecotones, highly contrasting or gradual habitat boundaries, plant species turnover etc.

NMDS analyses: Firstly, the NMDS plots should be part of the main manuscript, and not presented as supplementary materials, as these are central to their findings. More importantly, there is not enough detail on how the NMDS analyses were constructed and conducted. In studies such as these, it’s important to at least present the outputs for each NMDS axes (ordination scores) so that the readership (and reviewers) can assess what each axes represents and how the ordination points relate to those axes of environmental and/or temporal variability. Since the NMDS analyses is only briefly explained, and little detail provided regarding the axes (i.e. each NMDS component), its extremely difficult to evaluate the performance of the NMDS and the authors interpretation (and overall conclusion). In addition, there is not enough detail in the legends or labelling of both Figs s3a and s3b and the use of elliptical contours gives the impression of significant overlap in habitat preferences – particularly in the dry season, less so in the wet season. But this would be a mis-interpretation. I would revise the NMDS analyses significantly. Avoid drawing elliptical contours/boundaries around each species points, and instead use different colour symbols to represent each species. I suggest that the authors construct NMDS ordination showing pairwise biplots for the main axes of habitat variability for each season, and perhaps for both seasons combined, supplementing this with boxplots representing the distribution of non-metric multidimensional scaling ordination scores for fields with presence or absence of each species. To take the analyses further and provide the readership with a more robust analytical approach, I suggest they use a series of either GLMMs or GAMMs, with camera trap location as a random factor (or as a nested factor within habitat type) and each of the main NMDS axes scores as predictor variables to examine how the NMDS ordination habitat gradients influence the relative abundance (RAI) of their target species. Finally, I’m not sure Fig. 3 provides any substantial weight for this manuscript, and perhaps this should be moved to supplementary material, or even deleted overall.

Reviewer 2 Report

  1. This MS lacks the Simple Summary and the line numbers.
  2. Furthermore, a deep language polishing is required, as the MS is full of language and syntax flaws. I suggest the authors to send it to a native English speaker for language revision.
  3. Abstract: some parts of the abstract are written with a different font.
  4. What do you mean by “community coexistence”? Please, clarify.
  5. What is a “U-shape” distribution? Please, clarify.
  6. Introduction: the first sentence requires a citation.
  7. “Blue sheep Pseudois nayaur is a mountain ungulate endemic to the Pan-Himalaya area.” Please, add a citation.
  8. The end of the Introduction should include detailed aims and predictions, to put the MS in an hypothesis-driven context.
  9. Figure 1 is unclear, as lacking any geographical reference. Add an inset with a map of China and Yunnan.
  10. How many camera traps did you use? Did you rotate them across different stations? Were they active throughout the 24h cycle?
  11. In the Discussion, it should be clearly stated that your sampling was not conducted for throughout the year.
  12. Please, add a “Conclusion” paragraph as recommended by the journal.

Round 2

Reviewer 1 Report

The authors have made some improvements to their manuscript and I would like to thank them for their work and addressing some of my earlier comments on their first draft. I do have some additional comments to make based on their revisions and response letter:

Simple summary: The authors claim that the lower elevation population may be a separate species ecologically (see sixth line) or that this population may represent incipient speciation, but the authors have no evidence for this anywhere in their study e.g. they present no genetic data to support this statement. All they show are the elevational gradient associations  for two disjunct populations along their selected elevational gradient. Any reference to the lower elevation population constituting a separate species is simply not valid (not based on these data anyhow) and has to be removed from the manuscript. In addition, I would ask the authors to ask for additional help to improve the English language throughout this summary.

Abstract Lines 45-50: This statement (about different populations of a species undergoing differentiation) is far too speculative and I suggest that its deleted as there is no evidence to even suggest this. No historical data are available in terms of the possible causes of these two disjunct populations (let alone to show that they are on different evolutionary trajectories) and its more than reasonable to assume that other factors may have also contributed to the elevational range distribution e.g. eradication of the species at mid-elevations, habitat loss/degredation within part of the elevational range, simple adaptation of one population to foraging on an invasive species (which suggests that these populations may show quite a high degree of dietary plasticity – more so than previously suspected) or the result of a historical accidental/deliberate introduction. These are far more plausible speculations than their suggestion of invasive species niche evolution.

Introduction Lines 54-65: As per my original comments, I found again that there is simply not enough background to the causal factors that drive elevational ranges in species populations in montane environments. I did not find their response to my original comment satisfactory, and I strongly encourage the authors to dig deeper and utilise a number of studies on montane elevational ranges and ecotones (see papers by Terborgh, Kessler, Hertzog, Jankowski and references therein, and may others from Latin America) to significantly expand this first paragraph in their Introduction.  

Methods Lines 408-410: What are the average (mean) temperatures and precipitation levels in their study site?

Methods Lines 410-412: Are there any ecotones along this elevation gradient? These are hugely significant for driving patterns of species richness and beta-diversity along elevational gradients (many examples on this again from Andean elevational range studies).

Results Section 3.2 Community assemblage and NMDS analyses: To me this subsection should be subtitled patterns of community assemblage. I welcome the additional text and improvements the authors have made to how the NMDS analyses were conducted and also the vastly improved figures and their inclusion. However, there are still no details on what each NMDS axes represents, since the ordination scores are not shown as supplementary materials. To suggest that the NMDS axes have no practical meaning in the figure legend is erroneous as they should (or could) represent axes of environmental (elevational and habitat) variability (otherwise the whole analyses is redundant). Non-metric multidimensional scaling is a well-established technique that has been used extensively to identify ecological communities and gradients in taxa such as bird communities (e.g. Clough et al., 2009; Borges et al., 2016; Fazaa et al., 2017), but can also be used to determine environmental gradients (see Laurance, 1994). I strongly encourage the authors to use the resultant NMDS axis scores as a proxy for the elevational and habitat gradient associations of their focal species. Examination of the distribution of NMDS ordination scores for blue sheep and the other focal species could then reveal a much clearer differentiation in the axis scores for presence versus absence for their target ungulate species across both NMDS1 and NMDS2 (or perhaps NMDS3?) i.e. which species show a clear association for presence with negative scores whilst absences may be distributed over the entire range of NMDS axes scores. This would provide the quantitative measure this manuscript sorely needs i.e. causal factors of elevational range distribution patterns, rather than widely speculate on incipient speciation of thew lower elevational population (for which they have no evidence for). Including NMDS ordination axes scores as potential causal factors (or even as predictor variables) in GLMMs or even GAMMs (with either RAI or presence-absence records from camera trap locations as the dependent variable) would also add significant weight to this manuscript (as outlined in my original review).

NMDS Figure legends: These need to be refined. NMDS stress is a measure of the goodness of fit of the resultant ordination feature space as compared to the original data feature space, where zero is equal to perfect representation (see Boyra et al., 2004). The authors should also consider using pairwise biplots of the first two NMDS dimensions or perhaps include the third (but its difficult to recommend this as no ordination scores are presented) – these would then reveal any significant overlap between the elevational associations of their target populations and species. 

Results Line 924: What do you mean by ‘stable’ community structure?

Author Response

Point 1: Simple summary: The authors claim that the lower elevation population may be a separate species ecologically (see sixth line) or that this population may represent incipient speciation, but the authors have no evidence for this anywhere in their study e.g. they present no genetic data to support this statement. All they show are the elevational gradient associations for two disjunct populations along their selected elevational gradient. Any reference to the lower elevation population constituting a separate species is simply not valid (not based on these data anyhow) and has to be removed from the manuscript. In addition, I would ask the authors to ask for additional help to improve the English language throughout this summary.
Response 1:

A separate species identified based on genetic data is a biological species. Here, the ecological species, was identified by ecological niche. The lower elevation population feed on different food compared with higher elevation population. And in addition, we have observed body size difference between them. So, we think here the geographic separate populations described as differentiation ecological species is reasonable, especially in the context of discussion.

The English language has been improved again. So sorry that there were so many spelling mistakes.

Point 2: Abstract Lines 45-50: This statement (about different populations of a species undergoing differentiation) is far too speculative and I suggest that its deleted as there is no evidence to even suggest this. No historical data are available in terms of the possible causes of these two disjunct populations (let alone to show that they are on different evolutionary trajectories) and its more than reasonable to assume that other factors may have also contributed to the elevational range distribution e.g. eradication of the species at mid-elevations, habitat loss/degredation within part of the elevational range, simple adaptation of one population to foraging on an invasive species (which suggests that these populations may show quite a high degree of dietary plasticity – more so than previously suspected) or the result of a historical accidental/deliberate introduction. These are far more plausible speculations than their suggestion of invasive species niche evolution.

Response 2: We have deleted the statement “undergoing differentiation” in the abstract, we just discuss this in the discussion part. Other factors may have also contributed to the elevational range distribution of blue sheep seem impossible in these area, due to that other species shared habitat are all show a hump distribution pattern. High degree of dietary plasticity is a driving force behind the creation of separate ecological species. Previous studies have shown that fruit flies fed by different diets can differentiate into new species in just a few generations.

Point 3: Introduction Lines 54-65: As per my original comments, I found again that there is simply not enough background to the causal factors that drive elevational ranges in species populations in montane environments. I did not find their response to my original comment satisfactory, and I strongly encourage the authors to dig deeper and utilise a number of studies on montane elevational ranges and ecotones (see papers by Terborgh, Kessler, Hertzog, Jankowski and references therein, and may others from Latin America) to significantly expand this first paragraph in their Introduction.  

Response 3: We have carefully read the classical papers by Terborgh and Jankowski and other related professors. Elevational range boundary is not what we are focusing on in this study. Instead, the pattern of species abundance distribution, i.e.,continuous or discontinuous, unimodal or bimodal, is given more attention. And for the distribution pattern, we have summarized that most of biological species patterns were approximate unimodal and continuous along elevation (Terborgh 1971; Jankowski et al. 2013), based on which we proposed our hypothesis. In this paper, we did not pay much attention to the causal factors and not collected environmental factors, so we only briefly introduce the causal factors in Line 54-57 (papers by Terborgh 1971, Jankowski et al. 2013, and Wen et al. 2021 were added).

Terborgh, J. 1971. Distributions on environmental gradients: theory and a preliminary interpretation of distributional patterns in the avifauna of the Cordillera Vilcabamba, Peru. Ecology. 52: 23 – 40.

Jankowski, J.E., Londono, G.A., Robinson, S.K. & Chappell, M.A.(2013). Exploring the role of physiology and biotic interactions in determining elevational ranges of tropical animals. Ecography (Cop.), 36, 001–002.

Zhixin Wen, Anderson Feijó, Jilong Cheng, Yuanbao Du, Deyan Ge, Lin Xia, Qisen Yang, Explaining mammalian abundance and elevational range size with body mass and niche characteristics, Journal of Mammalogy,Volume 102,Issue 1,February 2021,Pages 13–27.

Point 4: Methods Lines 408-410: What are the average (mean) temperatures and precipitation levels in their study site?

Response 4: The annual average temperature ranges from 16.5℃ (2000 m) to–7.4℃ (5500 m), and the annual precipitation ranged from 200 mm (2000 m) to 1000 mm (4500 m) from low to high elevation in our study site. Those were added in Lines 111-113.

Point 5: Methods Lines 410-412: Are there any ecotones along this elevation gradient? These are hugely significant for driving patterns of species richness and beta-diversity along elevational gradients (many examples on this again from Andean elevational range studies).

Response 5: Yes, there are ecotones between junction vegetation types, spans around meters to dozens of meters elevational range. And species richness are higher in those ecotones. In this study we care about the population abundance distribution patterns at larger space scale. And based on Terborgh, J. 1971, ecotones could change population abundance or species richness rapidly, but not create bimodal distribution pattern. So ecotones were not given special consideration in this study.  

Point 6: Results Section 3.2 Community assemblage and NMDS analyses: To me this subsection should be subtitled patterns of community assemblage. I welcome the additional text and improvements the authors have made to how the NMDS analyses were conducted and also the vastly improved figures and their inclusion. However, there are still no details on what each NMDS axes represents, since the ordination scores are not shown as supplementary materials. To suggest that the NMDS axes have no practical meaning in the figure legend is erroneous as they should (or could) represent axes of environmental (elevational and habitat) variability (otherwise the whole analyses is redundant). Non-metric multidimensional scaling is a well-established technique that has been used extensively to identify ecological communities and gradients in taxa such as bird communities (e.g. Clough et al., 2009; Borges et al., 2016; Fazaa et al., 2017), but can also be used to determine environmental gradients (see Laurance, 1994). I strongly encourage the authors to use the resultant NMDS axis scores as a proxy for the elevational and habitat gradient associations of their focal species. Examination of the distribution of NMDS ordination scores for blue sheep and the other focal species could then reveal a much clearer differentiation in the axis scores for presence versus absence for their target ungulate species across both NMDS1 and NMDS2 (or perhaps NMDS3?) i.e. which species show a clear association for presence with negative scores whilst absences may be distributed over the entire range of NMDS axes scores. This would provide the quantitative measure this manuscript sorely needs i.e. causal factors of elevational range distribution patterns, rather than widely speculate on incipient speciation of thew lower elevational population (for which they have no evidence for). Including NMDS ordination axes scores as potential causal factors (or even as predictor variables) in GLMMs or even GAMMs (with either RAI or presence-absence records from camera trap locations as the dependent variable) would also add significant weight to this manuscript (as outlined in my original review).

Response 6:

We have subtitled this subsection as “patterns of community assemblage”.

How the NMDS analyses were conducted was supplied in Line 147-151 and 187-188.

The ordination scores by NMDS both for camera sample traps and species are shown in Supplementary Materials 5. We have examined the distribution of NMDS ordination scores for blue sheep and the other focal species in Supplementary Materials 4.

The GLMMs or GAMMs are not engaged because Figure 3 has shown the community difference among vegetation types (elevations), and we did not include any other environmental variables in our study.

Please give us more comments if we didn't answer your question clearly.

Point 7: NMDS Figure legends: These need to be refined. NMDS stress is a measure of the goodness of fit of the resultant ordination feature space as compared to the original data feature space, where zero is equal to perfect representation (see Boyra et al., 2004). The authors should also consider using pairwise biplots of the first two NMDS dimensions or perhaps include the third (but its difficult to recommend this as no ordination scores are presented) – these would then reveal any significant overlap between the elevational associations of their target populations and species. 

Response 7:

We have revised the Figure 3 legends. We have biplot the first two NMDS dimensions (two dimensions are reasonable because the stress value is 0.06) of target populations in Supplementary Materials 4. And the elevational (vagetation) associations of ungulate populations were consistence with what indicated in Figure 2.

Point 8: Results Line 924: What do you mean by ‘stable’ community structure?

Response 8:

It means “consistence”. We have changed the word.

Reviewer 2 Report

Authors have now amended the MS following all of my previous comments. Therefore, the paper is to be accepted (still, I do not see line numbers, indeed).

Best regards,

Emiliano Mori

Round 3

Reviewer 1 Report

I would like the thank the authors for their continued efforts to revise their manuscript. The overall language of the manuscript and figures are now much improved. I have some further comments on their submission:

Discussion: The final paragraph is still far too speculative and needs some substantial revision (perhaps it needs to be replaced entirely). Overall, there is too much essence of population differentiation in the Discussion and Introduction. The authors could end their Discussion section by simply calling for further studies, particularly those involving population genetics that could determine the level of differentiation between the two blue sheep populations, and maybe shed some light as to the source of the lower elevational population but without making claims that they are on different evolutionary trajectories toward species-level differentiation (the authors simply cannot suggest speciation without any indication of genetic divergence). This could be framed in a way that stresses that the causes (source) of these disjunct populations remain unknown, and may have arisen from a number of factors such as historical introductions/movements by people. (which is much more likely). There is no evidence to suggest that they have ever occurred along the entire regional elevational gradient – this point has to be made explicitly in their manuscript. Yes – the lower elevational population has become adapted to forage on an invasive cactus, but that can be simply attributed to local adaptation i.e. blue sheep are generalist feeders that can show local population level adaptation to feeding on specific food plants (and local adaptation can occur without incipient speciation - some avian examples of this).

What is also missing from the Discussion is an honest appraisal of their camera trapping method and the NMDS analyses. How well did their sampling protocol perform in recording the ungulate community and their target species? What about using detection probabilities to indicate camera trapping performance (this can be done simply by using N-Mixture models). What recommendations could the authors suggest for improved monitoring, or more detailed analyses of the species habitat preferences (i.e. inclusion of additional variables based on the previous mentioned literature)?

Supplementary materials S4 a and b: These are very good biplots and I would include these in the main manuscript. Note that each axes is mislabelled MDS rather than NMDS. Can the authors confirm that these are the same NMDS ordination axes as the community NMDS plots? If not, how different are these (and what do they represent)? Can the authors also highlight the significance of the dashed line at 0 on the NMDS2 axis in each plot (and if relevant, does this aid interpretation of the niche position of each species above/below it). Furthermore, the statement that the location of each species name represents its ecological niche implies that each NMDS axis represents an environmental gradient. These must be defined in their Results and Discussion (and associated figures) along with those in Figure 3.